# Stakeholders' perspectives on clinical trial acceptability and approach to consent within a limited timeframe: a mixed methods study

Elizabeth Deja ,[1] Chloe Donohue,[2] Malcolm G Semple,[3,4] Kerry Woolfall,[1] for the BESS Investigators

¹Department of Public Health, Policy and Systems, University of Liverpool, Liverpool, UK
²Liverpool Clinical Trials Centre, University of Liverpool, Liverpool, UK
³NIHR Health Protection Research Unit in Emerging and Zoonotic Infections, University of Liverpool, Liverpool, UK
⁴Respiratory Medicine, Alder Hey Children's Hospital, Liverpool, UK

**Correspondence to**
Dr Elizabeth Deja;
e.deja@liverpool.ac.uk

## ABSTRACT

**Objectives** The Bronchiolitis Endotracheal Surfactant Study (BESS) is a randomised controlled trial to determine the efficacy of endo-tracheal surfactant therapy for critically ill infants with bronchiolitis. To explore acceptability of BESS, including approach to consent within a limited time frame, we explored parent and staff experiences of trial involvement in the first two bronchiolitis seasons to inform subsequent trial conduct.

**Design** A mixed-method embedded study involving a site staff survey, questionnaires and interviews with parents approached about BESS.

**Setting** Fourteen UK paediatric intensive care units.

**Participants** Of the 179 parents of children approached to take part in BESS, 75 parents (of 69 children) took part in the embedded study. Of these, 55/69 (78%) completed a questionnaire, and 15/69 (21%) were interviewed. Thirty-eight staff completed a questionnaire.

**Results** Parents and staff found the trial acceptable. All constructs of the Adapted Theoretical Framework of Acceptability were met. Parents viewed surfactant as being low risk and hoped their child's participation would help others in the future. Although parents supported research without prior consent in studies of time critical interventions, they believed there was sufficient time to consider this trial. Parents recommended that prospective informed consent should continue to be sought for BESS. Many felt that the time between the consent process and intervention being administered took too long and should be 'streamlined' to avoid delays in administration of trial interventions. Staff described how the training and trial processes worked well, yet patients were missed due to lack of staff to deliver the intervention, particularly at weekends.

**Conclusion** Parents and staff supported BESS trial and highlighted aspects of the protocol, which should be refined, including a streamlined informed consent process. Findings will be useful to inform proportionate approaches to consent in future paediatric trials where there is a short timeframe for consent discussions.

**Trial registration number** ISRCTN11746266.

## INTRODUCTION

Bronchiolitis of infancy is a seasonal respiratory viral disease that most commonly causes

### STRENGTHS AND LIMITATIONS OF THIS STUDY

⇒ Use of the Deja *et al*'s (2021) adapted theoretical framework of acceptability allowed trial acceptability to be evaluated as a multifaceted construct as opposed to a poorly defined binary (acceptable/not acceptable) approach.

⇒ Acceptability was explored from multiple perspectives providing an in-depth understanding of key stakeholder views.

⇒ Acceptability was measured at multiple time points. The study may have benefited from more direct feedback from parents who declined their child's involvement in Bronchiolitis Endotracheal Surfactant Study.

rhinorrhoea, mild fever and a wet cough but in severe cases can result in feeding difficulties and respiratory distress. It is the single most common reason for hospital admission of infants (children age <1 year).[1] The youngest of these, those born prematurely and those with underlying conditions are most often and most severely affected.[2] There is no vaccine or specific treatment for bronchiolitis.[3] Despite advances in the provision of non-invasive modes of respiratory support, admissions to UK Paediatric Intensive Care Units (PICU) and duration of mechanical ventilation for life-threatening bronchiolitis have remained fairly static,[1] with the exception of during the COVID pandemic which indirectly lead to a decrease in all childhood infections.[4]

Pulmonary surfactant is secreted by lung alveolar cells to reduce surface tension so increasing compliance of the lungs, allowing them to inflate more easily and reduce the work of breathing.[5] Studies of infants with life-threatening bronchiolitis show reduced lung compliance and relative surfactant deficiency.[6–10] The systematic review by Jat and Chawla (2015) found that surfactant therapy

in infants with critical illness due to bronchiolitis was: safe, improved gas exchange and reduced both duration of mechanical ventilation and length of stay on PICU.[5]

The Bronchiolitis Endotracheal Surfactant Study (BESS) is a phase 2, blinded randomised controlled trial designed to explore the efficacy and mechanism of surfactant therapy compared with air placebo for critically ill infants with bronchiolitis (see online supplemental material 1 for protocol). BESS is recruiting in up to 14 PICUs in UK tertiary hospitals, with the intervention mostly being administered by respiratory physiotherapists. Informed consent was sought from parents/legal representatives (referred to hereon as parents) for their child's participation in BESS. Due to the time-dependent nature of the intervention, randomisation is required within 48 hours of intubation. This meant that urgent action was needed for the purposes of the trial, which is in line with legislation requirements for research without prior consent (RWPC).[6] However, it was unclear if informed consent could be reasonably sought within this limited time frame. In designing the study, we identified a need to explore parents' views on the acceptability of seeking informed consent in BESS within a limited time window with a view to consider an alternative approach, such as RWPC,[7] depending on parents' views and experiences of the recruitment and consent process.

Qualitative research to incorporate staff, patient or parent perspectives in the design of a clinical trial can help ensure the trial is acceptable, appropriate and possible to conduct.[8] BESS included an embedded mixed methods study in the first two seasons of the trial to explore staff and parents' perspectives on the acceptability of the trial; approach to recruitment and consent; decision-making in the emergency setting; and barriers to participation. This paper presents the findings from the embedded study to inform ongoing recruitment to BESS and future trials in emergency and critical care settings.

## METHODS
### Study design
A mixed methods study embedded within a clinical trial (BESS). Questionnaires and interviews with parents approached for informed consent in BESS, and a survey of BESS site staff involved in trial recruitment, consent procedures and intervention administration. We used previous embedded or trial feasibility studies[9–12] and the Adapted Theoretical Framework of Acceptability (ATFA)[13] to inform the design, including sample estimation, recruitment strategy, parent interview topic guide (see online supplemental material 2), parent questionnaire and site staff questionnaire (see online supplemental material 3). The parent questionnaire followed the same format as those used in similar studies[10 12 14] consisting of Likert scale questions from the Decision Making Control Instrument[15] and free text responses, taking on average 5 min to complete (see online supplemental material 4).

Preliminary findings from parent interviews in seasons 1 and 2 (2019/2020) also informed the development of the Site Staff Questionnaire, which consisted of 19 questions around the staffs' professional background, views on BESS training, BESS research process and BESS consent process. The site staff questionnaire was conducted at the end of season 2.

### Patient and public involvement
The BESS study and embedded study had patient and public involvement throughout. Six parents of infants admitted to PICU and ventilated for life-threatening bronchiolitis and one member of the NIHR GenerationR Alliance Young People's Advisory Group co-developed the study design and all study documents, some of which went on to be part of the BESS Parents Advisory Group. Two members are co-applicants and sit on the Trial Management Group and Trial Steering Committee. Representatives of the Paediatric Intensive Care Society as stakeholders in the research output reviewed an early draft and this submitted application.

### Parents/legal representatives
#### Eligibility, recruitment and sampling procedure
All parents of children approached about BESS were eligible to take part if they spoke English, including those who declined their child's participation in BESS.

Site staff provided parents with information about the embedded study as part of the BESS participant information sheet and recruitment discussions. Staff asked each parent (decliners and consenters) if they would like to complete the questionnaire after the recruitment discussion and/or if they were willing to take part in a telephone interview with the BESS researcher (ED) after their child had been discharged from hospital. Parents placed completed questionnaires in a stamped addressed envelope, which were collected by site staff and posted to ED. As part of the recruitment process, site research staff asked parents who declined to take part in BESS to provide a reason why consent was not provided. If a reason was provided, this was recorded on the BESS screening log.

Data collection continued until thematic saturation, where additional data did not lead to any new major themes identified during analysis. Researchers were also looking for high levels of 'information redundancy'[16–18] and information power, the point when data are deemed to address the study aims; sample specificity, such as experience relevant to the study aims, and sample diversity.[16 19]

#### Interview screening and conduct
Parents were contacted to arrange a telephone interview within one month of consent after checking each child's discharge and survival status with sites. Initially, parents were contacted in sequential order, then purposively sampled to ensure parents from all sites and decliners to BESS were represented, not just consenters from the first sites opened (the largest population). Informed consent

for audio recording of interviews was checked verbally at the time of the interview. The interviews explored views and experiences about: their child's admission to hospital, the BESS consent process, including how and when the trial was introduced by staff; information materials; and consent decision-making. This included questions about how trial processes could be improved and the potential use and acceptability of RWPC in subsequent seasons (see online supplemental material 2). Respondent validation was used to add unanticipated topics to the topic guide as interviewing and analysis progressed.[18] After the interview was complete, parents were sent a thank you letter, and a £20 Amazon voucher for their time.

### Site staff questionnaire

At the end of season 2, the BESS Trial Manager (CD) sent an email invitation to site staff requesting their participation in the online questionnaire and to cascade the link to colleagues involved in the trial. Email invitations described how completion of the questionnaire was taken as an indication of consent. Reminders were sent by (CD) and the BESS Chief Investigator (MGS) to all site key contacts after 1 week.

### Analysis

Digital audio recordings were transcribed verbatim by a professional transcription company (UK Transcription, Brighton) and anonymised. Qualitative analysis of interviews and open response questionnaire and screening log data was interpretive and iterative.[20] Using a thematic analysis approach, the aim was to provide an accurate representation of views on trial acceptability, design and processes[21] (see online supplemental material 5). NVivo V.12 software (QSR International Pty, Melbourne, Australia) was used to assist in the organisation and coding of data. Data from the parent and staff questionnaires were cleaned and entered into SPSS V.24.0 (IBM Corp.). Descriptive statistics are presented with percentages. Synthesis of qualitative and quantitative data drew on the constant comparative method.[22 23] This involved ED (PhD, female psychologist, Research Associate) and KW (PhD, female, social scientist, Reader) looking across quantitative and qualitative themes and quantitative output for themes/data output related to trial acceptability.

A final stage of analysis involved consideration of both qualitative and quantitative findings against the ATFA[13 24] to help conceptualise and discuss the overall acceptability of the proposed trial.

The ATFA is designed to assist researchers in assessing the acceptability of healthcare interventions including paediatric clinical trials. The framework explores eight aspects of acceptability at three possible time points, before (*prospective*), during (*concurrent*) and after (*retrospective*) the intervention (see online supplemental material 6). Our data were a mix of *Concurrent* and *Retrospective* data (see tables 1 and 2).

## RESULTS

### Sample

Of the 179 parents of children approached to take part in BESS, 75 parents (of 69 children) took part in the embedded study (see figure 1). Of these, 55/69 (78%) completed a questionnaire, 10/69 (15%) took part in an interview and 5/69 (7%) took part in both methods from 13/14 (93%) BESS sites. Thematic saturation was reached after eight interviews. An additional seven interviews were conducted to ensure variance across BESS sites and achieve information power.[16] Parents were interviewed on average (mean) 27 days (SD 7.8 days, range 17–43 days) after randomisation. The mean duration of PICU admission was 8 days (range 2.5–18 days), with a mean of 7 days (range 2.5–17 days) on a ventilator and a mean age of 70 days old (range 15–156 days). Interviews took on average (mean) 37 minutes (SD 9.6 minutes, range 20–57). Only 1/75 (1%) parent who consented to the embedded study had declined to consent to their child's involvement in BESS. Thirty-seven (37/44, 84%) parents provided site research staff with a reason why they had declined consent, which was recorded in the BESS screening log.

Thirty-nine staff at all 14 open sites BESS sites completed the online questionnaire including 14 doctors, 12 physiotherapists, 9 research nurses and 1 nurse. All participants apart from the site trial coordinator were involved in the clinical care of children with varied levels of experience of recruiting to paediatric clinical trials (range 0–20 years, median 4 years).

### Main findings

#### Acceptability of the BESS trial

Staff supported the trial, meeting all eight components of the ATFA (as shown in table 2). They had a clear understanding of the aim of the study and how to follow the protocol. They viewed the administration of the intervention as being acceptable and ethical with free text comments highlighting how the trial had been: 'well received by PICU staff and parents' (P31, Research nurse). Indeed, parents described their support for the trial. Many appeared to view BESS as an extension of standard care as the physiotherapy and tests conducted may have occurred anyway at different times in their child's clinical care. Consent decisions appeared to be informed (see below) as parents understood what BESS was investigating and why. They described their child's involvement as being low risk and were reassured by information about how surfactant was used in premature babies with very few side effects:

> 'I asked if there would be any side effects and things like that, and she said because it wasn't a new drug, she said there haven't been on the ones they had used it on previously, that they used it on premature babies.' P04, mother, interview.

Participation was not considered to place an additional burden on the child, although some parents found

**Table 1** Parent concordant acceptability of Bronchiolitis Endotracheal Surfactant Study (BESS) mapped to the adapted framework of acceptability Deja et al[13] 2021

| Affective attitude | Burden | Ethicality | Intervention coherence | Opportunity costs | Self-efficacy | Perceived effectiveness | Trust |
|---|---|---|---|---|---|---|---|
| 135/179 (75%) consented for their child to take part in BESS. | 50/60 (83%) did not find it difficult to take in information about BESS 47/60 (78%) did not find it difficult to make a decision about BESS "We felt the process was straightforward" MQ34 | 60/60 (100%) agreed that they were satisfied with the consent process 55/60 (92%) felt that medical studies like BESS are important | 60/60 (100%) agreed that the Information received about BESS was clear and straightforward to understand | No relevant data was reported at this time point | 60/67 (95%) felt that they made the decision for their child to take part in the pilot trial. 54/60 (90%) felt in control of the decision to take part. "Medical team made it clear that this decision was mine fully and I could withdraw consent at any time. This knowledge made participating much easier.' MQ45 | 50/53 (88.3%) Selected 'helping my child' as a reason for taking part 57/60=95% selected "to help other children in the future' as a reason for taking part "I think all children would benefit from this great work thanks" FQ42 | 45/60 (75%) selected "because I trusted the doctor or nurse who explained BESS' as a reason for taking part. "Trusting the doctor or nurse explaining BESS to parents is going to be an important deciding factor subconsciously for most parents" FQ30 |
| 9/179 declined to take part because they did not want their child to receive porcine surfactant 19/179 Do not want to take part in research | 'My partner & I decided it was too much to take on when our son is so sick, too much already going on.' MQ53- decliner' | | | | | | |

Tables key: shaded fields highlight potentially unacceptable aspects of the trial MQ mother open questionnaire response, MF: father open questionnaire response.
Note, not selecting this *did not mean* that they did not trust the staff, it was just not a identified reason for their consent.
FI, father interview; MI, mother interview.

**Table 2** Retrospective acceptability of BESS mapped to the adapted framework of acceptability Deja et al[13] 2021

| | Affective attitude | Burden | Ethicality | Intervention coherence | Opportunity costs | Self-efficacy | Perceived effectiveness | Trust |
|---|---|---|---|---|---|---|---|---|
| **Parental acceptability** | | | | | | | | |
| | Happy with trial "obviously I was approached for the BESS trial, which I was quite happy for him to take part in." MI02 | "But the standard of care for the placebo is just normal care anyway' MI06 | 'You feel powerless, to be entered into a research study without somebody talk to you about it you've got even more power taken away from you.' MI12  'you want the test to actually be effective if you're going to take part, don't you? You don't want to waste time.' FI11 | 'If they can do a trial that shows that actually giving the surfactant or not brings them out of it quicker' MI04  'literally any question I had in my head, this leaflet then had contained the answers' MI05 | 'There are not really risks, so there's not really any problem, all it could possibly do is produce positives. FI13 | 'She was so pleasant and she was so easy to talk to. Even when I wasn't discussing the paperwork and the trial with her, if I had any questions, I could easily walk up and ask, if I had any enquiries. She was very, very easy to approach.' MI02 | 'Then it was a realisation of, oh, my God, so you've just told me this thing that could make this difference and could get him off the ventilator quicker, and he might not get it anyway' MI05 | 'I don't think they'd put your babies in danger. They are here to get them better, so you just kind of trust in what they are doing.' MI4 |
| **Staff acceptability** | | | | | | | | |
| | Overall how acceptable do you find the BESS study? 20/36 (53%) very acceptable 18/36 (47%) acceptable "If it will have a clinical benefit which outweighs potential risk, then it is definitely acceptable" (P14) | "Low on personnel on the ground as medics are undertaking the intervention which takes considerable time in a busy unit" (P06, Doctor) | 10/36 (26%) found it very acceptable and 28/36 (74%) acceptable to administer the surfactant. 6/36 (16%) found it very acceptable and 31/36 (82%) found it acceptable to administer the placebo | The majority of staff rated the training as 'excellent' (27/36, 75%) or 'good' (9/36, 25%). | "We tried to time other interventions that required muscle relaxants for example, ETT tape changes with the time the placebo needed to be given" (P39, Physiotherapist). | 34/36 (87%) staff indicated that they had not experienced difficulties adhering to the BESS protocol. | 'I think it's important that we look at ways to improve care and treatment so research trials are vital. Reducing ventilator days would help free up precious ITU beds during winter and hopefully reduce problems with lung injuries' (P17) | Not mentioned in questionnaire responses |

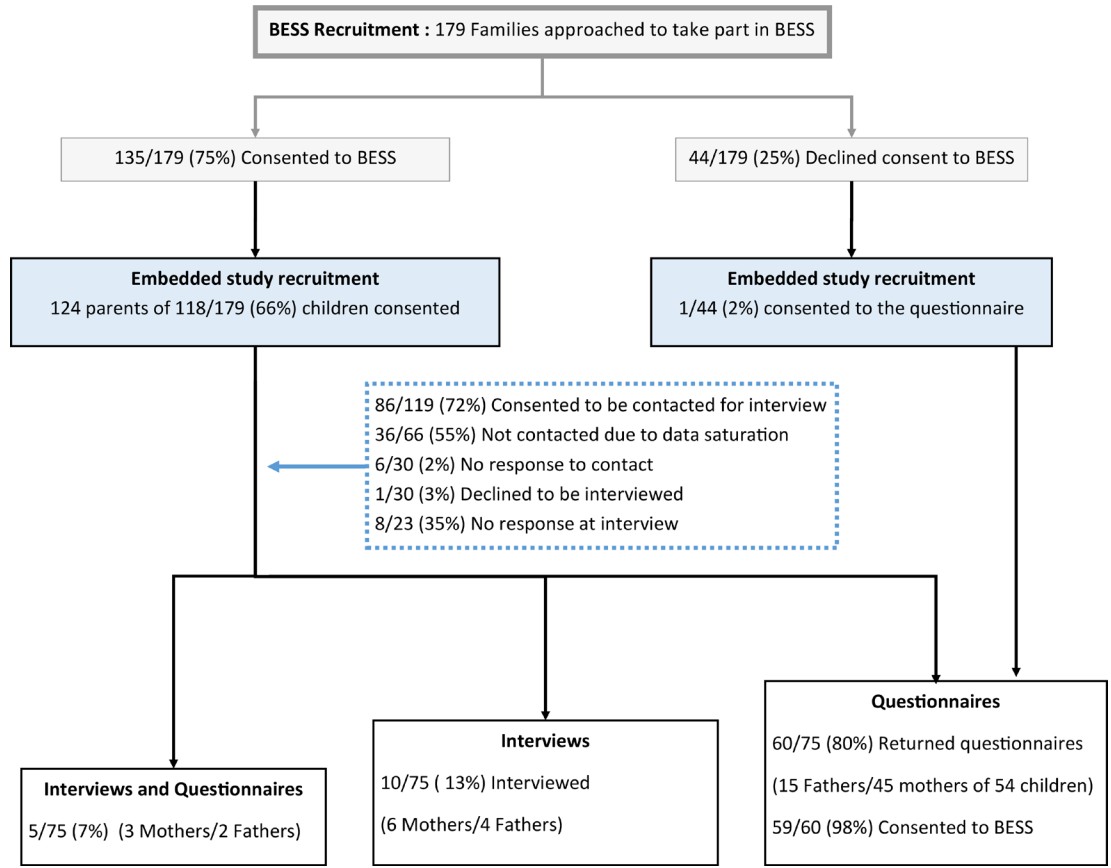

**Figure 1** Parent recruitment. BESS, Bronchiolitis Endotracheal Surfactant Study.

physiotherapy uncomfortable to watch and chose to leave the room:

> 'A lot of his physios, I was watching one where they'd, like, shake your baby. (Laughter) Then they push down on their ribs like they compress the ribs. I'm sure that that's what they were doing before they were going to possibly give the placebo or the surfactant. … It's frightening; I left… Because I thought, they don't need a panicky mum around them.' P14, mother, interview.

### Recruitment and consent in an emergency setting

Early in the interview, parents were asked to explain their understanding of the BESS Study. While the depth of descriptions varied, all gave a clear and accurate account of the purpose of the trial. Many outlined primary and secondary outcomes, including how surfactant might: 'reduce the amount of time on a ventilator' (P12, mother, interview), prevent: 'lungs from sticking' (P09, father, interview), help them: 'to expand and open their lungs' (P02, mother, interview) and hopefully help: 'them recover quicker' (P10, father, interview) therefore reducing parental and child distress, cost, ventilator-related infections.

Despite being approached soon after their child's admission to PICU, all interview and questionnaire participants stated that they were given sufficient time to consider the information and ask questions and that staff had broached BESS at 'an appropriate time' (100%, questionnaire item) (see tables 1 and 2). Parents described being able to make an informed consent decision about the trial, despite the highly emotive and stressful situation:

> 'I guess the main elephant in the room is, do I feel as if, because I was in a stressful situation, an emotional situation, where I wasn't in my right mind, where I'm having to deal with a lot of different, very scary, very emotional, very worrying situations, was I making clear and conscious decisions? The answer is yes, I was' P13, father, interview.

A total of 44/179 (25%) declined their child's participation in the BESS study. Reasons given for not taking part included: situational capacity as there was: 'too much already going on.' (Mother, questionnaire, decliner) and: 'do not want to take part in research' (19/179, 11%; Screening logs). Others declined as their child was close to coming off the ventilator at the time of final consent. A minority of parents 'did not want their child to receive surfactant' (9/179, 5%, Screening logs). However, most: 'families were much more open to surfactant than I expected,' (P19, Research nurse) 'especially if their child had been premature and had surfactant in the neonatal period.' (P10, Physiotherapist). One nurse noted that the number of decliners was similar to other PICU trials: 'few

exceptions where they didn't want child as a guinea pig, but no more than usual' (P6, Nurse).

### Facilitating quicker intervention delivery

Both groups highlighted that sometimes there were relatively long stretches of time between initial trial recruitment discussions and trial interventions beginning. The length of time was dependent on the availability of research nurses (who largely work 09:00–17:00 on weekdays) and parents for the consent discussions and availability of physiotherapists to administer the intervention: 'delaying the whole process' (P15, Research nurse):

'I think the only thing that could have possibly done better was doing it sooner because…potentially, it [the intervention] could have started a day earlier. The actual time between them approaching me, to me to find out what it is, to me to say yes and then actually starting the trial, I feel that could have been really streamlined' P13, father, interview.

'Low on personnel on the ground as medics are undertaking the intervention which takes considerable time in a busy unit' P06, Doctor

In addition to increasing the: 'potential for patients to begin weaning from the ventilator and therefore not be eligible' (P23, Research nurse). During interviews, parents questioned the ethicality and impact of delaying a potentially effective treatment they had consented to:

'If there was someone on hand to make it happen despite the shift change […] because you want the test to actually be effective if you're going to take part, don't you? You don't want to waste time.' P11, father, interview.

Parents provided practical suggestions for speeding up the recruitment and consent process, for example, having the written information available at the bedside so that they would likely have read it before the first approach allowing for those comfortable to consent without a follow-up discussion. Therefore, removing a step in the recruitment process timeline. Additionally, they highlighted the need to have more staff available that can take consent and administer the intervention.

### Perceptions of RWPC in BESS

Towards the end of the interviews, a definition of RWPC was read to parents (figure 2). All parents responded positively to the use of RWPC in critical care research in emergency situations for trials that were regarded as being low risk to their child's well-being.

'As long as it's something that doesn't have any risks associated with it, or there wouldn't be a reason why you wouldn't want them to do it and it's something that's going to help them, then I think it's (RWPC) a good thing.' P08, mother, interview.

Five parents described consenting to another trial that had used a RWPC approach during the same hospital admittance. Nevertheless, the majority (n=11/15, 74%) of parents interviewed suggested that BESS should continue to use an informed consent process as the intervention was not viewed to be time critical in that surfactant was regarded as something to hopefully get children: 'off the ventilator quicker' rather than an emergency intervention that would be lifesaving: 'if we don't give this to her immediately' (P08, mother, interview). In contrast, three parents were in favour of RWPC in the BESS study and there were no strong objections against its possible use, as the intervention was viewed to be low risk, would facilitate the intervention being given more quickly, thus avoiding missed patients and increasing recruitment rates, as the following quote illustrates:

'If you'd had just cracked on and done it, it could've made a difference quicker, but by the time we got around to the actual consent, he was being extubated' P05, mother, interview

### DISCUSSION

The aim of this study was to explore the acceptability of the BESS Study to help inform recruitment in subsequent bronchiolitis seasons and trials in other paediatric critical care settings. The ATFA[13] allowed us to explore the multifaceted construct of acceptability, with eight components to consider. Our data suggest that the BESS Study met all eight of the constructs. BESS

In BESS and most research, consent is obtained before children are entered into a study. In some trials conducted in emergency situations parents are asked to provide consent after their child had already been actively involved in the Trial. At this point consent is sought to continue in the trial. This is known as research without prior consent or deferred consent.

The reason that this legislation is in place is because in situations like this (ie, in paediatric critical care), there's a belief that there's no time to have a full discussion and time to reflect about the research and that actually having that discussion might delay treatment.

**Figure 2** Description of research without prior consent (RWPC) read to participants during interview.

was viewed by parents and PICU staff as understandable, posing low to no risk and having possible benefits to child participants.

Our findings suggest that both parents and staff viewed the BESS study as ethical and potentially beneficial to children with bronchiolitis. At the study design stage, there was uncertainty about whether an informed consent approach was appropriate for BESS or whether RWPC should be used due to the critical care context and the need for 'urgent action' for purposes of the trial, which is in line with RWPC legislation requirements.[25] Although RWPC was supported by a minority of parents as it would reduce any delay in their child receiving the intervention, many stated there was opportunity for an appropriately timed trial discussion before randomisation. Our findings support the continued use of a proportionate approach[26] to informed consent in this context. Parents stated that the length of the consent process and staff availability impacted the timeliness of administering the BESS intervention. They were concerned that such delays may reduce the potential effectiveness of the intervention. The nature of the intervention, which was viewed as being low risk to their child, meant that parents felt they could make an informed decision after one conversation with research staff, when their child's condition was stable. A streamlined consent process, along with other suggested changes to study processes should allow the full intervention to be administered in a timelier manner.

The seasonal nature of bronchiolitis provided an opportunity to review and adapt trial processes, increasing opportunities to make the trial more acceptable acceptability and feasibility to conduct. However, most trials do not have this ability leading to potential research and resource waste.[27] Our findings highlight the importance of pre-trial research or embedded studies in which the potential time windows between randomisation and intervention delivery can be clearly mapped and parent views on the most appropriate consent approach are fully explored.

The main strength of this study is that acceptability was measured at multiple time points, from different perspectives providing an in-depth understanding of key stakeholder views. The study may have benefited from more direct feedback from parents who declined their child's involvement in BESS.[28] We kept recruitment open for decliners for an additional season to try to address this gap in knowledge. However, their wish to 'not take part in research' and 'situational incapacity' in BESS, also applied to the embedded study and was compounded by low recruitment in season three as a consequence of COVID.[4] We believe that data from screening logs and the one decliner questionnaire provided insight into reasons for declining consent. In addition, the overall number of decliners was relatively low. Therefore, these missing data should not have a large impact on our findings.

## CONCLUSION

Our paper found BESS to be feasible and acceptable, supported the continued use of informed consent and highlighted areas for study process improvements. Future trials should consider proportionate consent processes when interventions are low risk and there is a short window for informed consent discussions.

**Acknowledgements** We firstly thank the children and families who participated in the BESS. Without your help this research could not have taken place. We also thank the principal investigators, respiratory physiotherapists and research nurses at each study site. We would also like to thank the National Institute for Health Research's Efficacy and Mechanism Evaluation (EME) Programme (ref: 12/205/28) for funding this research.

**Collaborators** BESS Investigators Chief Investigator: Malcolm G Semple PhD (Department of Clinical Infection, Microbiology and Immunology, Institute of Infection, Veterinary and Ecological Sciences, University of Liverpool and Department of Respiratory Medicine, Alder Hey Children's NHS Foundation Trust, Liverpool). Deputy Chief Investigator: Paul S McNamara PhD (Department of Women's and Children's Health, Institute of Life Course and Medical Sciences, University of Liverpool and Department of Respiratory Medicine, Alder Hey Children's NHS Foundation Trust, Liverpool). Coinvestigators: Evette Allen PhD, Clare Fowler HNC (Parent Lay Representatives), Catrin Barker PGDipClinPharm, Kentigern Thorburn MD, Matthew Peak PhD, Paul C Ritson MSc, Clare van Miert PhD (Alder Hey Children's NHS Foundation Trust, Liverpool), Ashley Best MSc, Chloe Donohue BSc, Ashley Jones PhD, Tracy Moitt, Laura Price BSc, Paula Williamson PhD (Liverpool Clinical Trials Centre, University of Liverpool, Liverpool), Howard W Clark PhD, Jens Madsen PhD (Elizabeth Garrett Anderson Institute for Women's Health, University College London, London), Anne Dawson, Colin Summers (Faculty of Health and Life Sciences, University of Liverpool, Liverpool), Elizabeth Deja PhD, Kerry Woolfall PhD (Institute of Population Health, University of Liverpool), Blessing Osaghae PhD, Mark Turner PhD (Department of Women's and Children's Health, Institute of Life Course and Medical Sciences, University of Liverpool), Madhuri Panchal BSc, Anthony Postle PhD (National Institute for Health Research Southampton Biomedical Research Centre Clinical and Experimental Sciences, Faculty of Medicine, University of Southampton, Southampton), John Pappachan MBBChir (Southampton Children's Hospital, Southampton), Roger Parslow PhD (Division of Epidemiology and Biostatistics, Leeds Institute of Cardiovascular and Metabolic Medicine, University of Leeds, Leeds), Jennifer Preston BA (NIHR Alder Hey Clinical Research Facility, University of Liverpool, Liverpool). Site Investigators: Alder Hey Children's Hospital, Liverpool: Kentigern Thorburn MD (Principal Investigator), Vanessa Compton, Peter Jirasek, Dawn Jones, Michael Mander, Laura O'Malley, Laura Rad, Laura Rimmer, Paul C Ritson MSc, Chris Simons. Birmingham Children's Hospital, Birmingham: Afeda Mohamed Ali MBBCh, Kevin P Morris MD (Principal Investigators), Cara Alexander BSc, Wendy Browne MSc, Hannah Child BSc, Stephanie Clarke BSc, Sarah Fox BN, Natalie Milburn BSc, Philip Milner FIBS, Samantha Owen BSc, Holly Parkin BSc, Harriet Payne BSc, Carly Tooke, Helen Winmill MSc. Bristol Royal Hospital for Children, Bristol: Peter J Davis MBChB (Principal Investigator), Katherine Baptiste, Sophie Coles BSc, Sarah-Jayne Eames BSc, Christina Linton BSc, Helen Marley BSc, Sarah Mogan BSc, Alvin Schadenberg PhD, John Stiven BSc. Great North Children's Hospital, Newcastle: Rachel Agbeko PhD (Principal Investigator), Rob Claydon PgDip, Christine Mackerness, Anna Stancombe BSc, Kate Teeley BSc, Grace Williamson MD. Leeds Children's Hospital, Leeds: Santosh Sundararajan MD (Principal Investigator), Ramesh Kumar, Kathryn Reeves BSc, Emily Scriven. Leicester Children's Hospital, Leicester: Raghu N Ramaiah MBBS (Principal Investigator), Rekha Patel. Nottingham Children's Hospital, Nottingham: Patrick E Davies BMBS (Principal Investigator), Lindsay Crate, Simon Gates. Oxford Children's Hospital, Oxford: James Weitz MBBS (Principal Investigator), Kirsten Beadon BSc, Rachel McMinnis BSc, Zoe Oliver BSc, Avishay Sarfatti BMBCh, Frances Sinfield, Charlotte Thompson. Royal Belfast Hospital for Sick Children, Belfast: Julie Richardson MBChB (Principal Investigator), Hilary Callaghan BSc, Vicki Linton BSc, Jeremy Lyons MD, Clara Nelson BSc, Mark Terris MBChB Royal Hospital for Children and Young People, Edinburgh: Tsz-Yan Milly Lo PhD (Principal Investigator), David Armstrong MD, Steve Cunningham PhD, Jackie McCormick, Andrea Wood MSc. Royal Hospital for Children, Glasgow: Richard Levin MBChB (Principal Investigator), Edgar Brincat MD, Christopher Lamb RN, Ross Marscheider. Royal Manchester Children's Hospital, Manchester: Stephen D Playfor DM (Principal Investigator), Bernadette C Gavin MSc, Dave J Morgan MSc, Lara T

Bunni MSc, Claire F Jennings MClinRes, Rebecca Marshall DipHE(Nur), Emma K Riley MNurSci. Southampton Children's Hospital, Southampton: John Pappachan MBBChir, Ahmed Osman MB ChB (Principal Investigators), Lorena Caruana, Hannah Clarke, Amber Cook BSc, Tracey Curtis, Nichola Etherington, Michael Griksaitis, Jenni McCorkell BSc, Christie Mellish, Simone Paulson BMBCh, Jenny Pond BSc, Catherine Postlethwaite DipPaedNur. Staffordshire Children's Hospital, Stoke: Constantinos Kanaris PhD (Principal Investigator), John Alexander MBChB, Nicola McClelland, Holly Minchin, Helen Parker, Pavanasam Ramesh, Joanne Tomlinson. The Royal London Children's Hospital, London: Simona Lampariello MD (Principal Investigator), Bessie Cipriano, Nosheen Khalid MSc, Ramiya Kirupananthan MBBS, Craig Knott MBBS, Tara Murray, Olivia Nugent, Nicolene Plaatjies, Samantha Reed, Natasha Roberts, Christa Ronan, Salman Siddiqi MB BS, Natasha Thorn BMBS.

**Contributors** MGS led the conception and design of the BESS study. KW designed and led the embedded study. CD facilitated the acquisition of data for analysis and trial management. ED conducted the interviews. ED and KW analysed and interpreted the data. ED and KW drafted the manuscript. All authors reviewed the manuscript critically for important intellectual content and provided approval of the version to be published. BESS Investigators' other contributions can be seen in the supplementary material. KW is the guarantor.

**Funding** This work was supported by the National Institute for Health Research's Efficacy and Mechanism Evaluation (EME) Programme (ref: 12/205/28).

**Disclaimer** This manuscript has been read and approved by all the authors, the requirements for authorship have been met, and each author believes that the manuscript represents honest work.

**Competing interests** MGS: Independent external and non-remunerated member of Pfizer's External Data Monitoring Committee for their mRNA vaccine program(s); HMG UK Scientific Advisory Group for Emergencies (SAGE), COVID-19 Response, Non-remunerated independent member March 2020 to March 2022. HMG UK New Emerging Respiratory Virus Threats Advisory Group (NERVTAG), Non-remunerated independent member since 2014; Integrum Scientific LLC, Greensboro, NC, USA, Chair of Infectious Disease Scientific Advisory Board; MedEx Solutions Ltd, Director; Integrum Scientific LLC, Greensboro, NC, USA, Minority Owner: MedEx Solutions Ltd, Majority Owner.

**Patient and public involvement** Patients and/or the public were involved in the design, or conduct, or reporting, or dissemination plans of this research. Refer to the Methods section for further details.

**Patient consent for publication** Not applicable.

**Ethics approval** Ethical approval for the study was given by the South Central Berkshire Research Ethics Committee (IRAS ID 220853). Participants gave informed consent to participate in the study before taking part.

**Provenance and peer review** Not commissioned; externally peer reviewed.

**Data availability statement** No data are available. The datasets generated during and/or analysed during the current qualitative study are not publicly available as consent was not sought for data sharing.

**ORCID iD**
Elizabeth Deja http://orcid.org/0000-0002-3626-4927

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
