## [Reviewer comments · BMJ Open]

ARTICLE DETAILS

TITLE (PROVISIONAL)	Stakeholders' perspectives on clinical trial acceptability and approach to consent within a limited timeframe: a mixed methods study.
AUTHORS	Deja, Elizabeth; Donohue, Chloe; Semple, Malcolm; Woolfall, Kerry; Investigators, BESS

VERSION 1 – REVIEW

REVIEWER	Ross, Catherine E. Boston Children's Hospital, Medical Critical Care
REVIEW RETURNED	24-Jul-2023

GENERAL COMMENTS	In “Parents and staff perspectives on the acceptability of The Bronchiolitis Endotracheal Surfactant Study (BESS): a mixed methods embedded study” the authors performed a mixed methods study within a pediatric critical care trial. The topic is important and interesting and the article presents novel qualitative data that may help inform the design of future pediatric critical care trials. Specifically, the description of family comprehension of the trial consent during an emotionally stressful period was very interesting and informative. However, I have several concerns about how the manuscript is organized and how the data are presented. In general, the methods and results need to be clearly divided and organized in a parallel manor to improve the readers' ability to follow. This may be improved in part by removing the data from the staff surveys which, to me, represent a far less important perspective on this topic. Additionally, the audience of BMJ Open (clinicians) needs to be considered in relation to how the qualitative methods are described and the tables presented. Finally, I believe the manuscript could be significantly shortened. TITLE: should include RWPC as this was the main question you were trying to answer ABSTRACT: -Spell out “38” at the beginning of the sentence INTRO: -Consider elaborating on why so many parents do not have the full 48 hours to decide about enrollment—are many patients transferred from outside facilities after intubation? Or is this an off hours / staffing issue? -I would like to see a better argument for why RWPC would be considered for BESS. If this is primarily a staffing issue, RWPC feels like a work around. METHODS: -Pg 5, line 11: I'm not sure what “Emended” means—did you mean “embedded”? -Pg 5, line 33: Would move this paragraph under the subheading for parents/LARs since it only pertains to them. Alternatively, since
--

	the methods are very focused on the family surveys/interviews, consider making the “Site Staff” it’s own separate section just before “Analysis” with a description of the survey they were given. It is not clear if the staff survey was the same as the parent survey. -Pg 5, line 35: The sentence beginning with “Information Power” is difficult to follow and seems to be incomplete. Are you trying to say that the subsequent items contributed to your Information Power? Also remember that the general BMJ Open audience may not be familiar with many of these phrases or concepts in qualitative research. -Pg 5, line 44: This seems like a description of how your sample size was determined—do you mean to say that you estimated that you would NEED this many patients to reach Information Power? I’m not sure its helpful to say what was estimated to happen. Also, would information power be limited to the qualitative interviews as opposed to surveys? -Pg 6, line 26 say “Author” be for “ED” or something else to indicate this is an author, or better yet make it a passive action (“parents were contacted to arrange . . .”) I initially interpreted this to mean the Emergency Department staff was contacting parents. -Was a primary investigator from BESS not involved in the coding process? -In general, the methods could be better organized and shortened to reduce the overall length of the manuscript. RESULTS: -All numericals at the beginning of a sentence should be spelled out -pg 7, line 39: change “at the close . . .” to “during the embedded study data collection period”. Also consider referring to the current study more specifically—i.e. instead of “embedded study” you could say “current study” or “current mixed methods study” or something along those lines. You would of course need to introduce this terminology early and keep it consistent throughout the manuscript. -pg 7, line 42: if 179 patients were approach, why is the denominator 119? Were all patients not asked about participating in the survey or did several of them not speak English? This information should be included in figure 1. -See notes on Figure 1 below, then eliminate most of the text from the first paragraph of the results which will be redundant with the figure. Instead, just refer to the figure. -The idea of “Thematic saturation” should be introduced in the methods section: i.e. “interviews were conducted until thematic saturation (explanation of thematic saturation) was achieved; additional interviews were conducted to ensure variance across BESS sites and achieve information power.” -pg 8, line 3: this paragraph could be organized better. The first sentence should be in the first paragraph. For this paragraph, start with a general description of the BESS patients (age, LOS stats, etc.). Then say when they were interviewed and how long it took. -In general, the data from surveys vs interviews needs to be presented separately. Clear organization / grouping with distinct subheadings would help. Currently the results discuss both intermixed and it is difficult to follow. -pg 8, line 15: Other than the first sentence, this paragraph should be included in methods, not results. -Pg 8, line 45: Do not start with Staff data. Keep this for the end of the results as a stand-alone section (if you choose to keep it at all. See below).
--	---

	-Pg 9, line 37: It is imperative to include here that parents were able to identify potential risks as were reviewed in the consent process to show these decisions were “informed”. -Table 2 should not be mentioned in the text before table 1. Reorder the text or the tables. -While I've included some comments above about how to organize the Staff contributions to this paper, I highly recommend removing this population entirely from this report. Overall, the main focus and by far the most important perspectives are those of the parents. While the Staff data have their place, I find that they detract from the focus of the paper and provide little data that is important or unique to the topic. -Similar to the methods section, the results section could be better organized and shortened to reduce the overall length of the manuscript. DISCUSSION: -Consider offering a comment on how to perform similar critical care studies whose interventions may not be “life-saving” or have limited but not emergent windows for enrollment. In this case, it appears to be better staffing during off hours, but there may be other, more generalizable ideas that were uncovered in your qualitative data. Need to introduce the concept in the discussion section before addressing it in the conclusion. TABLES AND FIGURES: -Figure 1: This should be a classic CONSORT diagram, starting with the number of patients approached to enroll in the BESS trial during the current study period and include here the proportion of people who participated in BESS vs those who didn't (perhaps separating these groups in 2 different boxes). Then list out the number of patients who were not asked to participate in the current study and reasons why (including non-English speakers or other reasons not asked). Then list the new denominator of people who agreed to participate and separate into those who were asked to do the survey, interview or both. Here you can list the reasons people did not complete the interviews. Eliminate information about timing (season 1/2), staff recruitment and parent interview details (the latter should be included in the text as mentioned above). -Figure 2: the description is incomplete and stops mid-sentence. This could be included as a supplemental material. Table 1: Shading needs to be explained in the footnote. Table 2: See above for the ordering of tables. -Tables in general: While I appreciate the approach of the presentation in the context of an established theoretical framework, this will not resonate with clinicians. I strongly recommend that Table 1 be a traditional table with each the quantitative items of the survey with summary statistics. Include participant characteristics (sex, race, relationship to the patient, etc.) These tables may still be included as supplemental tables, but in the main paper will be difficult for clinicians to understand.
--	--

REVIEWER	Blackwood, Bronagh Queen's University Belfast, Wellcome-Wolfson Institute for Experimental Medicine
REVIEW RETURNED	16-Sep-2023

GENERAL COMMENTS	Thank you for the opportunity to review this important study that focuses on using feedback from stakeholders in rolling out a trial. Not many investigators using these methods to strengthen trial implementation and as such the paper adds to trial methodology.
--

	There are some areas that I found unclear and I suggest comments that I feel would provide clarity and strengthen the paper.  1. Title – Could this be revisited. As written it intimates that the study encompasses the whole trial, rather than early findings from two seasons of the trial that focus on consent and early perceptions. 2. Introduction – many readers will not be familiar with the role of surfactant in bronchiolitis. Therefore provide some background information into this so that readers may better understand the choices parents had to make in consenting to the trial and to expect from the intervention. 3. The last paragraph of the introduction is very clear in defining the aims of the study – I feel the findings would be clearer if they reflected these aims. 4. Methods – there were many references to supplementary material although I was only provided with the trial protocol as the supplement. If the text is referring to the protocol as the sole supplement, then I feel it would be much clearer and easier for the reader if (a) the reader was directed to the particular page/s, or (b) a better choice would be to extract the relevant text and provide this in the supplement. 5. Questionnaires – this lacked methodological detail on construction, content, time to complete and piloting for understanding. 6. Likewise, interviews lacked detail on the interview schedule, duration, timing and place of interviews. More clarity is required to explain sampling – particularly around sequential recruitment and change to purposive recruitment. 7. Analysis - for interviews, it's unclear how the themes were generated, and this should be clearer in the results. E.g. demonstrated how the quotes corresponded to codes and how the codes were categorised into themes – in many cases this is shown in a table. 8. My main uncertainty in this paper, is why there was a need to use the Theoretical Framework of Acceptability in this preliminary study. In my view, the aims provided in the introduction are more apt and the findings would be clearer and stronger if reported this way. I felt the data were forced into the TFA – additionally it was not clear what was adapted in the TFA and why. For a reader who is unfamiliar with the TFA, Tables 1 and 2 will be confusing. 9. Main findings – it's not clear if the bold headings are meant to be the themes - they appear to be statements rather than themes. As stated earlier, I feel the findings would benefit from remaining close to the study aims. 10. The discussion was very good and clear. 11. Figure 1 would be clearer in a CONSORT type flowchart. 12. Figure 2 – I was unsure what this was meant to be – it was unfinished. 13. Some attention required to sentence structure and spelling in a few places
--	---

VERSION 1 – AUTHOR RESPONSE

--	--

Reviewer: 1 Dr. Catherine E. Ross, Boston Children's Hospital Comments to the Author: In "Parents and staff perspectives on the acceptability of The Bronchiolitis Endotracheal Surfactant Study (BESS): a mixed methods embedded study" the authors performed a mixed methods study within a pediatric critical care trial. The topic is important and interesting and the article presents novel qualitative data that may help inform the design of future pediatric critical care trials. Specifically, the description of family comprehension of the trial consent during an emotionally stressful period was very interesting and informative. However, I have several concerns about how the manuscript is organized and how the data are presented. In general, the methods and results need to be clearly divided and organized in a parallel manor to improve the readers' ability to follow. This may be improved in part by removing the data from the staff surveys which, to me, represent a far less important perspective on this topic. Additionally, the audience of BMJ Open (clinicians) needs to be considered in relation to how the qualitative methods are described and the tables presented. Finally, I believe the manuscript could be significantly shortened.	Thank you for your thoughtful and detailed comments. Each of your concerns have been addressed in turn. Please see below. We hope this results in a much clearer paper.
1 TITLE: should include RWPC as this was the main question you were trying to answer	Adjusted (to also include reviewer two's comment)
2 ABSTRACT: -Spell out "38" at the beginning of the sentence	Changed. We have also changed other sentences in the abstracts starting with numerical numbers
3 INTRO: 3.1 Consider elaborating on why so many parents do not have the full 48 hours to decide about enrollment—are many patients transferred from outside facilities after intubation? Or is this an off hours / staffing issue?	We have removed that line as it wasn't clear, this is was not about staffing. We have added how urgent action was needed for the purposes of the trial, which fits with legislation on RWPC
3.2 I would like to see a better argument for why RWPC would be considered for BESS. If this is primarily a staffing issue, RWPC feels like a work around.	This had been added. RWPC was considered as BESS met the legislation requirements for RWPC (e.g. urgent action needed for purposes of trial, treatment needed to be given urgently) yet we were uncertain if it was reasonably practicable (or acceptable) to obtain consent within such a limited time window. We have clarified this uncertainty, which was the

	rationale for the embedded study.
METHODS: 4. Pg 5, line 11: I'm not sure what "Emended" means—did you mean "embedded"?	Correct and changed
4. 1 Pg 5, line 33: Would move this paragraph under the subheading for parents/LARs since it only pertains to them. Alternatively, since the methods are very focused on the family surveys/interviews, consider making the "Site Staff" it's own separate section just before "Analysis" with a description of the survey they were given. It is not clear if the staff survey was the same as the parent survey.	Site staff section moved to before analysis as suggested and additional detail added to clarify.
4.2 Pg 5, line 35: The sentence beginning with "Information Power" is difficult to follow and seems to be incomplete. Are you trying to say that the subsequent items contributed to your Information Power? Also remember that the general BMJ Open audience may not be familiar with many of these phrases or concepts in qualitative research. 4.3 Pg 5, line 44: This seems like a description of how your sample size was determined—do you mean to say that you estimated that you would NEED this many patients to reach Information Power? I'm not sure its helpful to say what was estimated to happen. Also, would information power be limited to the qualitative interviews as opposed to surveys?	4.2 and 4.3 were addressed by changing the location and wording of the 'information power' explanation within the methods section.
-Pg 6, line 26 say "Author" be for "ED" or something else to indicate this is an author, or better yet make it a passive action ("parents were contacted to arrange . . .") I initially interpreted this to mean the Emergency Department staff was contacting parents.	This has been changed to 'passive action' for clarity
-Was a primary investigator from BESS not involved in the coding process?	No The qualitative lead (KW) was involved in the coding process. The Primary investigator of BESS is not a qualitative researcher and this was outside of his role.
In general, the methods could be better organized and shortened to reduce the overall length of the manuscript.	Based on your feedback and that of reviewer two the methods section has been re arranged and shortened
RESULTS: -All numericals at the beginning of a sentence should be spelled out	Amended as requested
Also consider referring to the current study more specifically—i.e. instead of "embedded study" you could say "current study" or	The terminology 'embedded study' refers to a study within a

“current mixed methods study” or something along those lines. You would of course need to introduce this terminology early and keep it consistent throughout the manuscript.	trial. This is a specific and recognised methodological approach. We have tweaked the title and methods to help clarify that this is the study design.
-pg 7, line 42: if 179 patients were approach, why is the denominator 119? Were all patients not asked about participating in the survey or did several of them not speak English? This information should be included in figure 1.	This section has been removed from the text however “125 parents of 119/179 (67 %) children consented for embedded study questionnaires and/or interviews “ has now been added to the new CONSORT diagram to clarify where the 119 denominator came from as this explanation was missing.
See notes on Figure 1 below, then eliminate most of the text from the first paragraph of the results which will be redundant with the figure. Instead, just refer to the figure	Text has been removed and new figure is referenced instead.
-pg 8, line 3: this paragraph could be organized better. The first sentence should be in the first paragraph. For this paragraph, start with a general description of the BESS patients (age, LOS stats, etc.). Then say when they were interviewed and how long it took.	This has been addressed in the other changes to the section.
-In general, the data from surveys vs interviews needs to be presented separately. Clear organization / grouping with distinct subheadings would help. Currently the results discuss both intermixed and it is difficult to follow.	More detail has been added to the analysis section to explain why the data is presented together. The organization / subheadings have been changes to improve clarity
-pg 8, line 15: Other than the first sentence, this paragraph should be included in methods, not results.	No longer applicable due to other changes to the manuscript.

-Pg 8, line 45: Do not start with Staff data. Keep this for the end of the results as a stand-alone section (if you choose to keep it at all. See below).	The opening of this section has been changed. However the Staff data is still presented first as it provided the context for the parental information.
-Pg 9, line 37: It is imperative to include here that parents were able to identify potential risks as were reviewed in the consent process to show these decisions were “informed”.	Understanding of the trial and associated risks was assessed during interviews to help establish if consent decisions were informed. We have added a line in the to clarify just before our description of parents’ views on potential side effects. Please see page 10.
-Table 2 should not be mentioned in the text before table 1. Reorder the text or the tables.	Tables one and two are presented for the first time in the analysis section in the correct order. The referral to table two in the results section is its second mention. We considered reordering the tables, however as the concordant acceptability comes before the retrospective acceptability the order and mentions are correct.
-While I’ve included some comments above about how to organize the Staff contributions to this paper, I highly recommend removing this population entirely from this report. Overall, the main focus and by far the most important perspectives are those of the parents. While the Staff data have their place, I find that they detract from the focus of the paper and provide little data that is important or unique to the topic.	While we agree that the parents’ perspectives are the most interesting within this paper. The acceptability of the BESS trial can only be fully established by looking at all stakeholders. We would like to keep staff perspectives in the paper to reflect the research undertaken and per the protocol agreed by Funder and REC(IRB).
-Similar to the methods section, the results section could be better organized and shortened to reduce the overall length of the manuscript.	We have reduced the length of the quotes to shorten the findings and changed the organisation linked to yours and reviewer 2’s comments.

TABLES AND FIGURES: -Figure 1: This should be a classic CONSORT diagram, starting with the number of patients approached to enroll in the BESS trial during the current study period and include here the proportion of people who participated in BESS vs those who didn't (perhaps separating these groups in 2 different boxes). Then list out the number of patients who were not asked to participate in the current study and reasons why (including non-English speakers or other reasons not asked). Then list the new denominator of people who agreed to participate and separate into those who were asked to do the survey, interview or both. Here you can list the reasons people did not complete the interviews. Eliminate information about timing (season 1/2), staff recruitment and parent interview details (the latter should be included in the text as mentioned above).	Changed to CONSORT diagram.
Figure 2: the description is incomplete and stops mid-sentence. This could be included as a supplemental material.	This has been amended to include all the text
Table 1: Shading needs to be explained in the footnote.	Added to the Key
-Tables in general: While I appreciate the approach of the presentation in the context of an established theoretical framework, this will not resonate with clinicians. I strongly recommend that Table 1 be a traditional table with each the quantitative items of the survey with summary statistics. Include participant characteristics (sex, race, relationship to the patient, etc.) These tables may still be included as supplemental tables, but in the main paper will be difficult for clinicians to understand.	Having reflected on this and other comments around the methods and the use of the ATFA our justification for the use of the framework was not clear. Accordingly we have adjusted the text within the manuscript to justify the use of the framework. The tables should be easier for non- qualitative researchers to understand. These mixed methods tables are needed within the main text to provide evidence of how our data support the conclusion that the BESS study is acceptable according the TFA criteria.
Reviewer: 2 Prof. Bronagh Blackwood, Queen's University Belfast	Thank you for your considered comments. We have addressed your points and agree that the

Comments to the Author: Thank you for the opportunity to review this important study that focuses on using feedback from stakeholders in rolling out a trial. Not many investigators using these methods to strengthen trial implementation and as such the paper adds to trial methodology. There are some areas that I found unclear and I suggest comments that I feel would provide clarity and strengthen the paper.	amendments add clarity and strengthen the paper.
1. Title – Could this be revisited. As written it intimates that the study encompasses the whole trial, rather than early findings from two seasons of the trial that focus on consent and early perceptions.	The title has been changed to refer to the consent focus. We have clarified the timing of this work in the abstract as this was difficult to add to the title without it becoming long winded and clunky
2. Introduction – many readers will not be familiar with the role of surfactant in bronchiolitis. Therefore provide some background information into this so that readers may better understand the choices parents had to make in consenting to the trial and to expect from the intervention	Information on surfactant has been added
3. The last paragraph of the introduction is very clear in defining the aims of the study – I feel the findings would be clearer if they reflected these aims.	The findings have now been presented to clearer reflect the aims
4. Methods – there were many references to supplementary material although I was only provided with the trial protocol as the supplement. If the text is referring to the protocol as the sole supplement, then I feel it would be much clearer and easier for the reader if (a) the reader was directed to the particular page/s, or (b) a better choice would be to extract the relevant text and provide this in the supplement.	There is now separate “supplementary material” documents in addition to the protocol.
5. Questionnaires – this lacked methodological detail on construction, content, time to complete and piloting for understanding.	Additional information has been added on the questionnaires
6. Likewise, interviews lacked detail on the - interview schedule,	The interview schedule is described within the methods but this had been better sign posed for clarity

- duration, timing - place of interviews. More clarity s required to explain sampling – particularly around sequential recruitment and change to purposive recruitment.	Duration/timing is stated with information on conducted interviews We have clarified that the interviews were conducted over telephone. Information added to explain the switch in sampling.
7. Analysis - for interviews, its unclear how the themes were generated, and this should be clearer in the results. E.g. demonstrated how the quotes corresponded to codes and how the codes were categorised into themes – in many cases this is shown in a table	Additional information on Analysis has been included and the requested table has been added to the supplementary materials due to word length restrictions.
8. My main uncertainty in this paper, is why there was a need to use the Theoretical Framework of Acceptability in this preliminary study. In my view, the aims provided in the introduction are more apt and the findings would be clearer and stronger if reported this way. I felt the data were forced into the TFA – additionally it was not clear what was adapted in the TFA and why. For a reader who is unfamiliar with the TFA, Tables 1 and 2 will be confusing.	There are multiple definitions of acceptability that often lack detail (all of which are included within the framework). The ATFA was built in to the feasibility study from the onset to define acceptability and demonstrate how we assessed it using our data. We feel this is an important part of the work to show rigour. However, we had not explained this within the paper and have now clarified this approach in the methods section and new layout of the findings
9. Main findings – it's not clear if the bold headings are meant to be the themes - they appear to be statements rather than themes. As	The findings have been re labelled to better reflect the

stated earlier, I feel the findings would benefit from remaining close to the study aims.	themes (related to study aims) rather than statement format
10. The discussion was very good and clear.	Thank you
11. Figure 1 would be clearer in a CONSORT type flowchart.	Figure 1 has been changed to a consort follow chart.
12. Figure 2 – I was unsure what this was meant to be – it was unfinished.	This had been corrected to include the missing text
13. Some attention required to sentence structure and spelling in a few places	This has been amended

VERSION 2 – REVIEW

REVIEWER	Ross, Catherine E. Boston Children's Hospital, Medical Critical Care
REVIEW RETURNED	31-Oct-2023

GENERAL COMMENTS	In this revision, the newly titled “Stakeholders’ perspectives on clinical trial acceptability and approach to consent within a limited timeframe: a mixed methods study” the authors report findings from their mixed methods study embedded within a pediatric critical care trial. The authors did an excellent job of addressing the critiques from the original submission. The manuscript is better organized and focused. Some minor errors remain: RESULTS:  -The phrase “75 parents of 69/179 children . . . “ is a bit confusing. For families that had more than one participant, did they provide the same response together or did each parent provide a separate response? If the former, consider rephrasing to “Of the 179 children recruited to BESS, 69 (38%) families participated in the embedded study, including 63 individual parents and 6 families with 2 parents contributing to the response.” The number of sites that participated can be stated elsewhere in this paragraph. Please also adjust this sentence in the abstract as well. -Eliminate 2nd paragraph and add the following to the preceding paragraph: “Only 1/69 (%) participants in the embedded study had declined to consent to the main trial.” □ this would be easier to follow with an appropriate CONSORT diagram, see notes on figures. -pg 10 line 47: “stated that THEY were given” -The number of patients who declined BESS here (19) does not add up to the number in your CONSORT diagram (179-135=44). -The methods do not describe how data was collected from BESS decliners who did not participate in the embedded study. You have a few sentences about it inappropriately in the results section. I also don’t think it is appropriate to include the staff’s speculations on why families declined participation. Please also explain the case report form in more detail (or consider omitting this section of the manuscript entirely as it does not address your primary question). TABLES AND FIGURES:  -Figure 1: While I appreciate the effort to convert to a classic CONSORT diagram, the flow of the current diagram is inconsistent and is difficult to follow.
--

	First Row: Box 1: number of patients approached to enroll in the BESS trial Second Row: Box 2: number of pts who consented; Box 3 number who did not consent Third Row: Box 3: number of consented BESS pts who consented to embedded study; Box 4: number of BESS decliners who consented to embedded study 4th row: Box 5, 6 & 7 (Interviews, Questionnaires, Both). For these include ONLY the final number who completed each. Between the 3rd and 4th rows, make a line leading to a separate box that shows the reasons the rest of the patients were lost, i.e. not contacted for interview, no response, etc. Each pt should only be counted once and all 179 should be accounted for.
--	--

VERSION 2 – AUTHOR RESPONSE

Reviewer: 1 comments	
3 In this revision, the newly titled “Stakeholders’ perspectives on clinical trial acceptability and approach to consent within a limited timeframe: a mixed methods study” the authors report findings from their mixed methods study embedded within a pediatric critical care trial. The authors did an excellent job of addressing the critiques from the original submission. The manuscript is better organized and focused. Some minor errors remain:	Dear Catherine thank you for taking the time to look at our article a second time and identify further points that needed addressing to improve clarity and content.
4 The phrase “75 parents of 69/179 children . . . “ is a bit confusing. For families that had more than one participant, did they provide the same response together or did each parent provide a separate response? If the former, consider rephrasing to “Of the 179 children recruited to BESS, 69 (38%) families participated in the embedded study, including 63 individual parents and 6 families with 2 parents contributing to the response.” The number of sites that participated can be stated elsewhere in this paragraph. Please also adjust this sentence in the abstract as well	We have changed the wording to “Of the 179 parents of children approached to take part in Bess, 75 parents (of 69 children) took part in the embedded study.” in both the methods and abstract moving the number of sires elsewhere. As this change put the abstract over the word count, minor edits were made to reduce the words back to 300.
5 Eliminate 2nd paragraph and add the following to the preceding paragraph: “Only 1/69 (%) participants in the embedded study had declined to consent to the main trial.” ♦ this would be easier to follow with an appropriate CONSORT diagram, see notes on figures.	This section has been amended as suggested.
6 pg 10 line 47: “stated that THEY were given”	‘They’ added

7 The number of patients who declined BESS here (19) does not add up to the number in your CONSORT diagram (179-135=44).	This has been corrected
8.1 The methods do not describe how data was collected from BESS decliners who did not participate in the embedded study. 8.2 You have a few sentences about it inappropriately in the results section. 8.3 I also don't think it is appropriate to include the staff's speculations on why families declined participation. 8.4 Please also explain the case report form in more detail (or consider omitting this section of the manuscript entirely as it does not address your primary question).	Decliners' data was collected in the same manner as consenters, we have tweaked the wording in the "Eligibility, recruitment and sampling procedure" to make this clearer. We have moved this to the methods section and expanded to address 8.4. We have kept the screening log data and removed the staff quote on why one family declined consent. We have added more detail about the case report form/ Bess screening log (the case report form summarised the screening log. for simplicity / clarity this has been changed to screening log in the paper) to show that this information comes from the decliners directly and is not conjecture as this was not clear before. We have also added this to the data analysis section. We would like to include this section it as it provides insight into acceptability of BESS and the consent process, both of which are aims of the paper.
9 TABLES AND FIGURES: Figure 1: While I appreciate the effort to convert to a classic CONSORT diagram, the flow of the current diagram is inconsistent and is difficult to follow. First Row: Box 1: number of patients approached to enroll in the BESS trial Second Row: Box 2: number of pts who consented; Box 3 number who did not consent Third Row: Box 3: number of consented BESS pts who consented to embedded study; Box 4: number of	Amended as suggested

BESS decliners who consented to embedded study
4th row: Box 5, 6 & 7 (Interviews, Questionnaires,
Both). For these include ONLY the final number who
completed each. Between the 3rd and 4th rows, make
a line leading to a separate box that shows the
reasons the rest of the patients were lost, i.e. not
contacted for interview, no response, etc. Each pt
should only be counted once and all 179 should be
accounted for.